# Exploration of the Corrosion Behavior of Electroless Plated Ni-P Amorphous Alloys via X-ray Photoelectron Spectroscopy

**DOI:** 10.3390/molecules28010377

**Published:** 2023-01-02

**Authors:** Zhizhen Li, Chaoqun Bian, Lingxia Hu

**Affiliations:** Pharmaceutical and Material Engineering School, Jinhua Polytechnic, Jinhua 321000, China

**Keywords:** Ni-P, electroless plating, crystallization process, XPS, anodic dissolution

## Abstract

A Ni-P amorphous alloy was deposited on a low carbon steel substrate via electroless plating. Further, the prepared samples were crystallized under the high temperature with a range from 200 °C to 500 °C in air for 1 h. The crystallization process was studied via XRD, AFM, and XPS, and anodic electrochemical behavior was investigated by potentiostatic methods in a 3.5 wt% NaCl solution. The experimental results indicate that the diffusion, dissolution, and enrichment of the component elements in the Ni-P alloy are essential during crystallization because the various corrosion behaviors corresponding to Ni and P are directly affected. More importantly, under the 400 °C treatment, H_2_PO_2^−^_ was enriched in the alloy, which effectively hinders the anodic dissolution of nickel and forms a complete adsorption layer on the surface of the alloy. Our results demonstrate that P can effectively block the anodic dissolution of Ni during the corrosion process, and the crystallization process can effectively promote the surface enrichment of P to improve the corrosion resistance of the coating.

## 1. Introduction

The corrosion resistance of amorphous electroless Ni-P alloy coatings is related not only to the properties of the plating bath [1,2,3] and plating conditions [4,5], but also to the elemental content [6,7,8] and post-treatment processing [9,10,11]. Coupled with the different measurement methods and conditions [12,13,14,15,16], an accurate and comprehensive description of the coating’s corrosion behavior can be difficult [17,18,19].

Numerous studies [20,21,22,23,24] have shown that the different corrosion mechanisms of nano-amorphous electroless coatings are caused by the differences in their crystallization processes. Xia et al. [25,26] studied the corrosion resistance of amorphous, as-plated nanocrystalline, and tempered nanocrystalline Ni-P—amorphous Ni-P had the best corrosion resistance. Crobu et al. [27,28] attributed the findings to the formation of passivation films on the corroded surface—film formation is mainly controlled by diffusion. The diffusion of solute atoms in nanocrystals is faster than that of coarse grains, and the surface of nanocrystals has many grain boundaries, which are conducive to the nucleation of passivation films.

Our previous research [29] demonstrated that the nanocrystalline structure of the coating alloy underwent changes (amorphous changes, the adjustment of amorphous changes, crystalline state changes) during crystallization of the alloy coating. The corrosion behavior of the coating concurrently changed from a uniform corrosion controlled by charge transfer to a uniform corrosion controlled by diffusion-grain boundary corrosions; this change impacted the corrosion resistance.

The corrosion resistance of the electroless plated Ni-P alloy is also affected by the products generated during corrosion. Flis et al. [30] showed that Ni-P alloys with low P content will form a thin oxide film with poor protection in neutral solution. The surface of the alloys with a high P content form Ni_3_(PO_4_)_2_·8H_2_O that inhibits the dissolution of the alloy. Zhao et al. [31] reported that the Ni-P alloy surface film is dominated by Ni(OH)_2_ with good protection under the open circuit potential, and low polarization potential. The resulting surface film is mainly composed of Ni_3_(PO_4_)_2_ with poor protection at high polarization potential. Bozzini et al. [32] suggest that the corrosion resistance of the Ni-P alloys is related to defects in the NiO passivation layer. Although the layer obtained under high anodic polarization contains phosphate, this is not a factor that affects the corrosion resistance of the coating. Diegle et al. [33,34] found that the Ni-20P in H_2_SO_4_ did not form a typical nickel oxide film at passivation potentials. Rather, the passivation process was controlled by the formation and adsorption of H_2_PO_2^−^_. This led to a P-enrichment phenomenon. Elsener et al. [35] also confirmed that there is no NiO passive film; rather, the corrosion resistance is due to the elemental P suppressing the dissolution of Ni via a diffusion mechanism.

In this report, the Ni-P amorphous alloy was successfully synthesized on a low carbon steel substrate by a simple fabrication process. Moreover, the crystallization process of Ni-P alloy was investigated via XRD, AFM, and XPS, and anodic electrochemical behavior was assessed by potentiostatic methods in a 3.5 wt% NaCl solution. It was evaluated that the diffusion, dissolution, and enrichment of the component elements in the Ni-P alloy during crystallization, and the various corrosion behaviors corresponding to Ni and P. Importantly, H_2_PO_2^−^_ was enriched in the alloy under the 400 °C treatment, which effectively hinders the anodic dissolution of nickel and forms a complete adsorption layer on the surface of the alloy. This work demonstrate that P can effectively block the anodic dissolution of Ni during the corrosion process, and the crystallization process can effectively promote the surface enrichment of P to improve the corrosion resistance of the coating.

## 2. Results and Discussion

### 2.1. Analysis of the Coating Composition and Microstructure

The amorphous structure of Ni-P thin film was determined by XRD (Figure 1). The diffraction pattern of the as-plated Ni-P sample has no peak except for the broad amorphous peak at 2θ = 45°, thus indicating the completely amorphous character of the as-plated sample, the results agreed with the studies from some other investigators [36,37]. After the alloy was heated to 200 °C and 300 °C, the intensity and position of the diffraction peak of the coating did not change considerably versus the as-plated state, thus indicating that the crystal structure was in an adjusting state. However, the crystallization was unclear. The appearance of various sharp diffraction peaks in the following XRD patterns indicated crystallization at high temperature. Figure 1 (200 °C to 400 °C) shows that both the number and the strength of those diffraction peaks increased with increasing heating temperature, indicating that the extent of the crystallization of the Ni-P amorphous alloy gradually increased. However, the XRD pattern no longer changed when treating the samples above 500 °C suggesting that the crystallization process was complete. Two kinds of crystalline diffraction peaks corresponding to Ni and Ni_3_P were observed simultaneously in the XRD patterns indicating that both Ni and Ni_3_P were formed at the same time during crystallization. However, it was reported that the position and strength of Ni_12_P_5_ and Ni_5_P_2_ were similar to those of Ni_3_P, and their crystal structures were easily converted into Ni_3_P. Therefore, the existence of Ni_12_P_5_ and Ni_5_P_2_ cannot be completely excluded.

Two-dimensional AFM images of the Ni-P samples heated at different temperatures with a scanning area of 20 µm × 20 µm are shown in Figure 2. As shown, all of the crystalline grains can be observed, and are found to be round with clear grain boundaries between them. During the heat treatment, the crystal cells became visible, flat, uniform in size, and the grain boundaries were narrower because the heat treatment reduced the porosity and internal stress of the crystal cells, and improved the flatness and density of the coating. The morphology of the coating heated at 300 °C is not significantly different from that treated at 200 °C, but the flatness of the coating increased, as shown by the chromaticity scale in the Figure. When the heat treatment temperature reached up to 400 °C, the cellular particle size still did not change much, but the particle surface changed significantly. There are many fine “particles” on the surface that are evenly and densely distributed on the cell particles, as in the formation of a new surface layer.

### 2.2. Tafel Curve of the Coating

Figure 3 shows the polarization curves of the alloy coatings in 3.5 wt% NaCl medium treated under the different temperatures. The corrosion potential of the sample showed a substantial positive shift after heat treatment and corrosion resistance was enhanced. The corrosion potential was the largest after 400 °C heat treatment, and the anodic polarization curve showed several states. It was in an active state from 0 to 100 mV. Between 200 mV and 500 mV, it was in a passivation state. The current density did not increase with the positive shift of electrode potential over this range, but it was not lower than the current density before passivation. When the potential was higher than 550 mV, the alloy entered an over-passivation region: The current density increased rapidly with increasing potential. However, the dissolution reaction was suppressed when the potential exceeded 650 mV, and the slope of the curve increased, which may be due to the deposition of corrosion products on the surface of the alloy. The good corrosion resistance of the alloy coating is because heat treatment can effectively eliminate the internal stress of the Ni-P alloy coating and reduce the porosity of the alloy. At the same time, the Ni phase and the Ni_3_P phase crystallize and form a dense nanocrystalline layer on the surface of the Ni-P alloy, as shown by AFM characterization. This leads to an effective barrier for the coating. This result has been confirmed by our group and Bai et al. [36,38].

### 2.3. XPS Analysis of the Coating

#### 2.3.1. XPS Analysis of Amorphous Ni-P Alloy Coating

The elemental composition of the surface and deep layer of electroless-plated Ni-P alloys were determined by XPS. Figure 4 shows the XPS and XPS-peak-splitting spectra of Ni 2p and P 2p peaks of the surface of as-plated Ni-P coating etched for 30 s. The surface of electroless Ni-P coating mainly contained Ni, P, C, and O; C and O came from environmental contamination. Figure 4a shows the XPS spectrum of Ni 2p. The Ni 2p peaks include 2p3/2 and 2p1/2 main peaks; there were corresponding secondary peaks near the main peak. According to the binding energy, the existing states of Ni could be determined as Ni^0^ and Ni^2+^, and the chemical states were NiO and Ni(OH)_2_. Figure 4c shows the XPS-peak-splitting processed spectrum of Ni 2p; the results are shown in Table 1. The Ni^0^ content was 58.23%, and the NiO content was 19.08%. The Ni(OH)_2_ content was 22.69%. The 0-valent Ni levels on the surface of the as-plated coating were relatively small and mainly in the form of Ni^2+^. Some of the Ni^2+^ came from the NiO that was produced by the oxidization of 0-valent Ni metal on the surface of the coating in air. The other part came from Ni^2+^ in the plating solution that hydrolyzed to form Ni(OH)_2_, which was adsorbed on the surface of the coating. Figure 4b,d show the XPS and XPS-peak-splitting processed spectra of P 2p, respectively (see more details in Table 1). According to the binding energy of P, the P existed as P^0^ and P^+^. The P elements mainly included P^0^ with binding energies of 129.4 eV and 130.4 eV and total content of 65.59% and P^+^(H_2_PO_2^−^_) with a binding energy of 132.7 eV and a content of 34.41%. These results suggest that the surface of the as-plated coating was composed of P elements reduced from the deposition process. The P^+^(H_2_PO_2^−^_) adsorbed on the surface of the coating participated in a reduction reaction.

XPS spectra were collected after the coating was etched for 30 s. The Ni 2p and P 2p peaks in the inner layer are shown in Figure 4e,f, respectively. The intensity of the main peak of Ni increased considerably, and the peaks at 856.1 eV and 874.0 eV disappeared, thus suggesting that the inside of the alloy coating mainly existed in the form of Ni^0^. The P had similar behavior and existed as a 0-valent element with binding energies of 129.4 eV and 129.9 eV.

#### 2.3.2. XPS Analysis of Amorphous Ni-P Alloy Coating during Crystallization

Figure 5 shows the XPS spectra of Ni 2p and P 2p peaks in the outer layer of the coating after the amorphous Ni-P alloy coating was subjected to heat treatments at the different temperatures. Versus those in the as-plated state, the intensity of Ni peaks near the binding energy of 855 eV and 872 eV increased dramatically after heat treatment at 200 °C. These correspond to the chemical states of NiO and Ni_2_O_3_ of Ni elements, indicating that the degree of oxidation of the Ni on the surface of the coating was intensified at higher temperature. After heat treatment at 300 °C, the spectrum of the alloy coating changed, and the main peak of Ni 2p3/2 shifted to a high binding energy by about 2.0 eV. The spectrum did not change substantially with further increases in the heat treatment temperature. The peak intensity of high-valent P (H_2_PO_4^−^_) at a binding energy of 133.22 eV increased after heat treatment to 200 °C; the P signal weakened for heat treatments at 300 °C and above. The P element likely diffused into the coating with high-temperature heat treatment. A film of Ni oxides and salts might also form on the surface of the coating at high temperatures thus preventing P from being detected.

Figure 6 shows the peak separation and processing for the spectra of Ni in the alloy coated via 400 °C heat treatment as well as P in the alloy coated at 200 °C (Table 2). Figure 6a shows that the total content of Ni^2+^ was 81.14%, and the content of Ni^3+^ was 18.86%. After high-temperature heat treatment, the 0-valent Ni on the surface of the coating did not exist, and Ni mainly existed in the form of oxides or oxygen-containing salts. Figure 6b shows that the valence state of the P did not change, but the high-temperature heat treatment reduced the P^0^ and P^1+^ contents while the content of P^5+^(H_2_PO_4^−^_) increased significantly. Some reports [35] have suggested that this is because P will be oxidized to P_4_O_6_ at room temperature, and a lone pair of electrons on each P atom in the P_4_O_6_ molecule can continue to be oxidized. High-temperature crystallization promotes the further oxidation of P to P_4_O_10_ leading to a strong moisture absorption property. This reacts with H_2_O in the environment to form H_3_PO_4_ during storage. XPS-peak-splitting data suggested that the total concentration of P increased after crystallization for surface enrichment. Therefore, after heat treatment at 300 °C, the failure to detect the P element on the surface of the alloy coating is caused by the coverage of Ni oxides or oxygen-containing salts generated during the crystallization process.

In contrast to the crystallization process of XRD analysis, the elements in the alloy coating did not change in valency during crystallization, indicating that the change in the crystal microstructure occurred during the crystallization process rather than via changes in the existing form of constituent elements. The grains in the coating were uniform during crystallization, which improved the corrosion resistance of the coating. This result suggests that the uniformity of the coating structure is an important factor affecting the corrosion resistance of the alloy coating as also confirmed by Habazaki et al. [39,40].

#### 2.3.3. XPS Analysis of Corrosion Product Film of Amorphous Ni-P Alloy Coating

Electrochemical corrosion is a dynamic process. When using XPS for elemental analysis of the corrosion product films of samples with different degrees of crystallization, it is impossible to combine the XPS test device with the electrochemical corrosion device. The sample in the ground state and the sample treated at 400 °C were selected for corrosion analysis. XPS was performed on the surface layer of the corroded part and the inner layer after etching, for 30 s (Figure 7 and Figure 8).

Figure 7a shows the peak intensity of Ni in the as-plated alloy coating before and after corrosion. The value was noticeably different. The peak intensity corresponding to Ni oxide decreased indicating that the corrosion process was accompanied by the dissolution of NiO or Ni_2_O_3_. The peak position and intensity of the P before and after corrosion did not change substantially, and P^0^ with a binding energy of about 129.3 eV was dominant (Figure 7c). After etching for 30 s, the Ni and P elements were still in the same state as the internal state of the original ground-state alloy coating (Figure 7c,d). The Ni in the alloy coated at 400 °C had the same change in Ni in the ground-state alloy coating before and after corrosion (Figure 8a,b). However, the P element that disappeared after high-temperature heat treatment appeared after corrosion as shown in Figure 8c, indicating that the dissolution rate of P was slower than that of Ni during corrosion. Chen et al. [41] also confirmed this view when studying the semiconductor properties of Ni-P coated alloy. Rather than P in the ground-state alloy coating, P existed as P^+^(H_2_PO_2^−^_) with a binding energy of about 132.7 eV. After etching for 30 s (Figure 8d), the peak position of the P shifted to a high binding energy showing a mixture of P^+^(H_2_PO_2^−^_) and P^5+^(H_2_PO_4^−^_) with binding energies around 132.5–133.5 eV.

Figure 9 shows the XPS-peak-splitting processed spectra of Ni 2p of the corroded alloy coatings at the ground state and with heat treatment at 400 °C (Table 3). Compared with the standard reference spectra, the existing states of Ni in the two alloy coatings were Ni^0^ and Ni^2+^(Ni(OH)_2_), and the corresponding nickel oxide was not detected. The Ni(OH)_2_ content was concurrently much higher than the ground-state alloy coating. When the Ni-P alloy is in its natural state, it is mainly composed of elemental NiO formed by Ni in air and a small amount of Ni(OH)_2_ (Figure 3). Different from the results obtained by Zhao et al. [31,34], the oxides of Ni are dissolved to form Ni(OH)_2_ when the alloy coating is in a corrosion-active state. However, Ni^0^ and Ni(OH)_2_ co-exist in the coating because Ni(OH)_2_ is easily dissolved in the corrosive medium and the presence of metallic Ni allows the alloy to be dissolved while increasing electrode potential.

Furthermore, the Ni, Ni(OH)_2_, and P were found in the corrosion products of the ground-state alloy coating. No corresponding nickel oxide or phosphate groups were found. The corrosion products of the alloy coated at 400 °C were Ni, Ni(OH)_2_, and H_2_PO_2^−^_. Combined with the polarization curve characteristics of the crystallization process of the alloy coating, these results suggest that the anodic dissolution of Ni can be effectively blocked during the corrosion process. The corrosion resistance of the alloy coating is thus improved. This feature is because the dissolution rate of the P is slower than Ni, and heat treatment can enrich the P on the surface of the alloy. The results further show that P alone cannot form a passivation film, or that its blocking effect is insufficient to inhibit the anodic dissolution at relatively high potentials. The H_2_PO_2^−^_ was enriched in the alloy coating with heat treatment at 400 °C, indicating that H_2_PO_2^−^_ effectively hinders the anodic dissolution of nickel on the surface of the alloy and forms a complete adsorption layer on the surface of the alloy. Costa et al. [33,42,43] also confirmed this view when studying the corrosion resistance of Ni-P coated alloy. This layer forms a physical barrier and generate a passivation film that greatly inhibits the anodic dissolution of Ni, leading to good corrosion resistance.

## 3. Materials and Methods

The substrate of the electroless plated sample was made of 45 steel (20 mm × 20 mm × 1.5 mm) following the published process flow and composition of the plating solution [44]. The steel substrate was immersed in the plating solution, and the volume of the plating solution was 1000 mL; the pH value of the plating solution was 4.5–5.0, the temperature was 88–90 °C, and the plating was performed for 3.5 h.

The thickness of the Ni-P coating was 35 μm, and the chemical composition of the Ni-P amorphous alloy was Ni_89.46_P_10.54_ using energy dispersive spectroscopy (EDS;LEO-438-VP, JEOL, Japan). The Ni-P alloy coatings were treated in air at the different temperatures (200 °C, 300 °C, 400 °C, and 500 °C) for 1 h. The amorphous structure of the alloy coating was determined by X-ray diffraction (XRD; KY-2000, Bruker AXS, Germany). In addition, the surface morphology of the alloy coating was characterized by atomic force microscopy (AFM; SII-SPI-3800N, SEIKO, Japan).

The electrochemical corrosion performance was tested with a PS-16A electrochemical test system. The exposed area of the test was 100 mm^2^, and other parts were sealed with insulating paint. The reference, counter, and sample electrodes were a calomel electrode, a platinum electrode, and the working electrode, respectively. The scan speed was 2 mV·s^−1^, and the corrosive solution was 3.5 wt.% NaCl solution. The experiment was performed at room temperature.

Electrochemical corrosion is a dynamic process. The chemical states of Ni and P atoms in the Ni-P alloy coatings with different states of samples, and different degrees of corrosion, were compared and analyzed by a Nanoscope IIIA X-ray photoelectron spectroscopy (XPS; Nanoscope IIIa, Digital Instruments, America) system. There was a charging effect in the experiment, and the binding energy needed to be calibrated to 284.80 eV using C1s as a standard.

## 4. Conclusions

In summary, a facile fabrication process was used to synthesize the Ni-P amorphous alloy on a low carbon steel substrate. The fabricated sample was crystallized under various temperatures from 200 °C to 500 °C for 1 h under air atmosphere. Moreover, importantly, the XRD, AFM, and XPS were employed to investigate the crystallization process, and the anodic electrochemical behavior was investigated by potentiostatic methods in a 3.5 wt% NaCl solution. Here, we summarize the main points of our results as follows:The change in the crystal microstructure mainly occurs in the crystallization process of the alloy coating rather than a change in the chemical state of the constituent elements. The uniformity of the alloy coating structure is an important factor affecting its corrosion resistance.P can effectively block the anodic dissolution of Ni during the corrosion process, and the crystallization process can effectively promote the surface enrichment of P to improve the corrosion resistance of the coating.P alone cannot form a passivation film at high polarization potentials. H_2_PO_2^−^_ species were enriched in the alloy coated at 400 °C. This effectively hinders the dissolution of nickel anode on the surface of the alloy and forms a complete adsorption layer on the surface of the alloy, thus generating a passivation film with good corrosion resistance.NiO was not detected on the polarized Ni-P alloys consistent with previous studies. Thus, the formation of an “oxide-type” passivation film can be excluded.

## Figures and Tables

**Figure 1 molecules-28-00377-f001:**
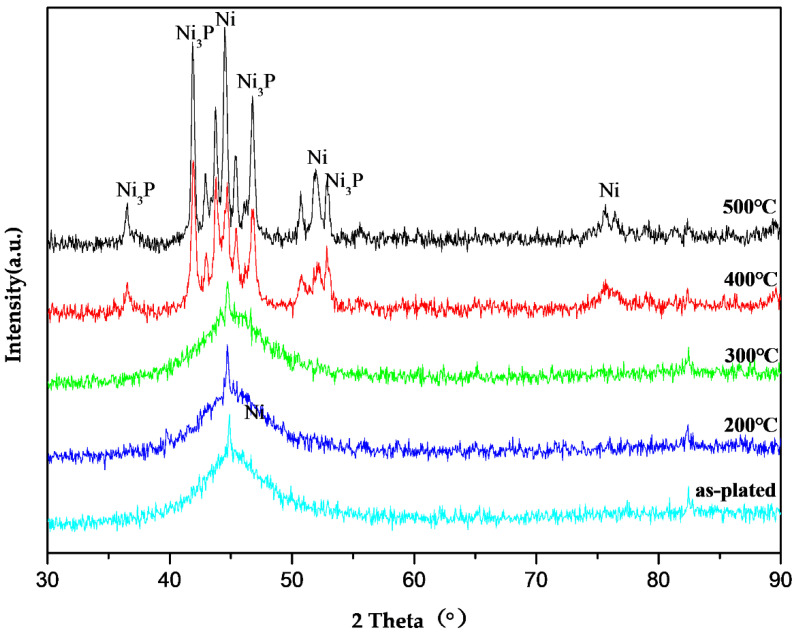
XRD patterns of Ni-P coating treated under different temperatures.

**Figure 2 molecules-28-00377-f002:**
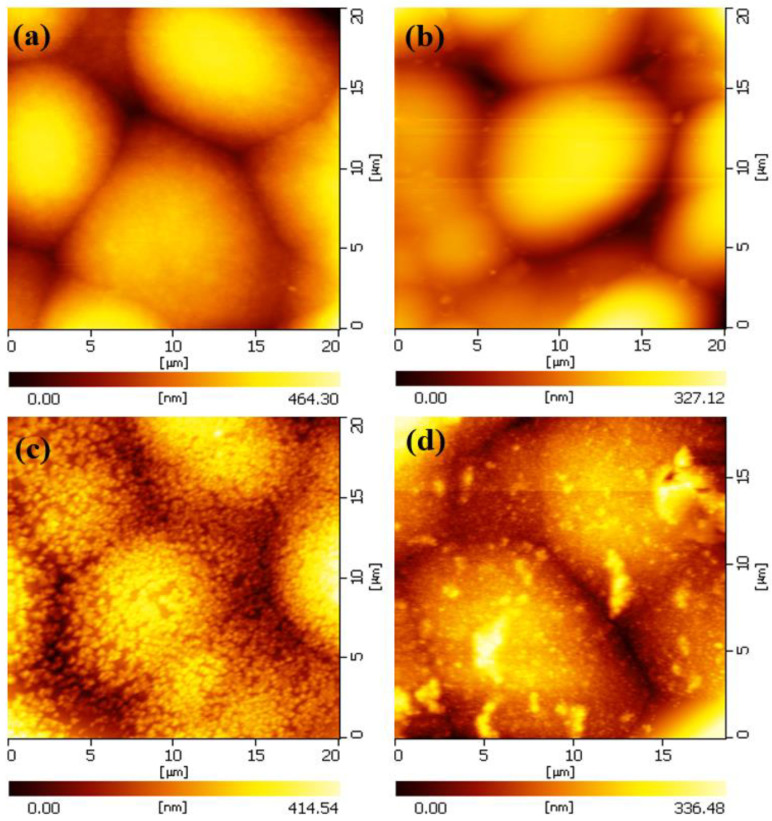
Two-dimensional AFM image of Ni-P coating after heated treatment: (**a**) 200 °C; (**b**) 300 °C; (**c**) 400 °C; (**d**) 500 °C.

**Figure 3 molecules-28-00377-f003:**
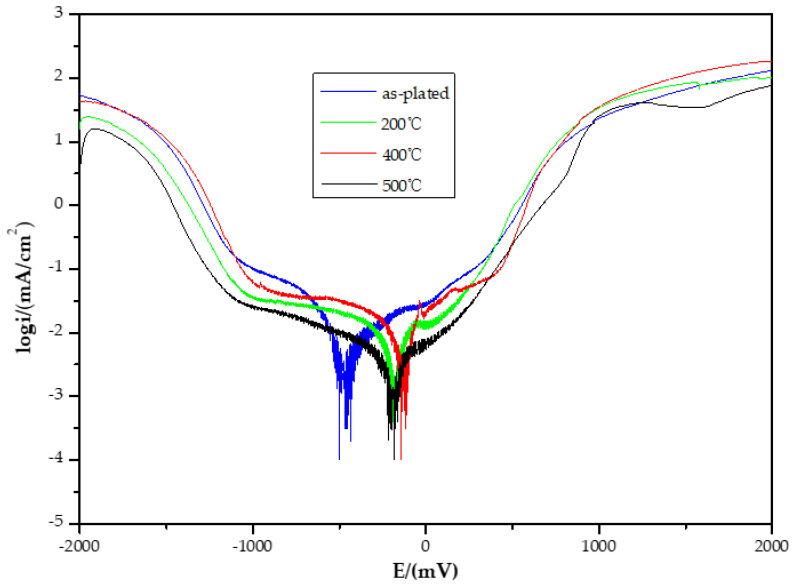
Polarization curves of Ni-P coatings with heat treatments at different temperatures.

**Figure 4 molecules-28-00377-f004:**
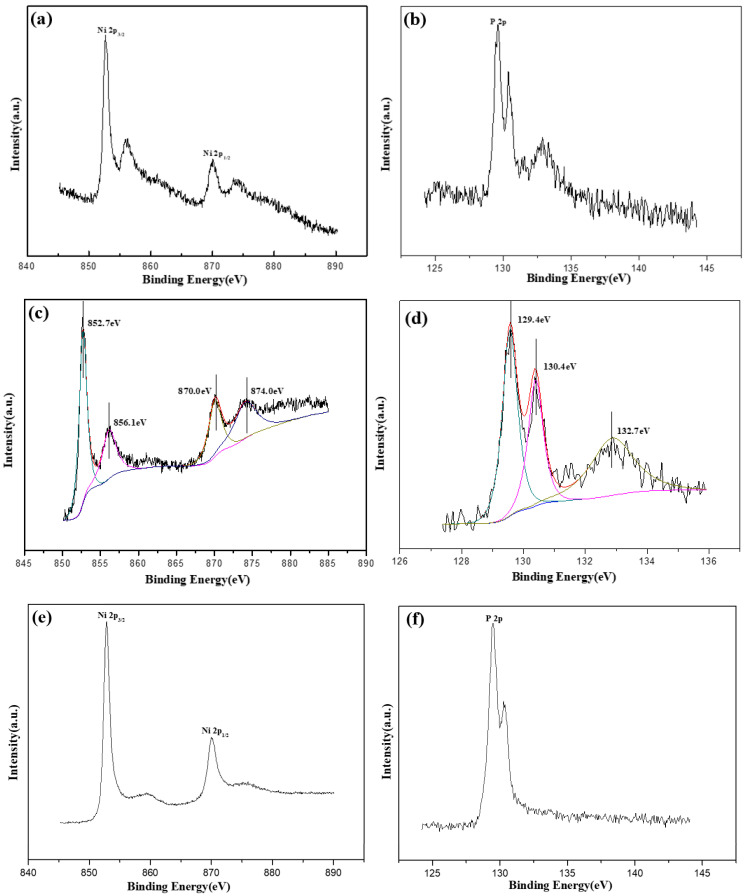
XPS analysis of Ni and P in electroless plated Ni-P coating: (**a**) XPS spectrum of Ni, (**b**) XPS spectrum of P, (**c**) peak-slitting processed spectrum of Ni, (**d**) peak-slitting processed spectrum of P, (**e**) XPS spectrum of Ni after etching for 30 s, and (**f**) XPS spectrum of P after etching for 30 s.

**Figure 5 molecules-28-00377-f005:**
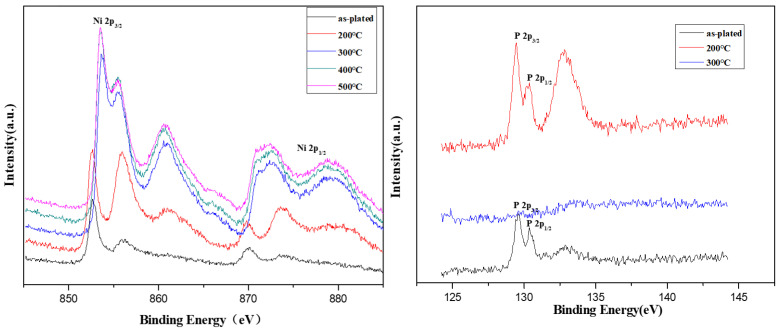
XPS analysis of Ni and P of the coatings with heat treatments at different temperatures.

**Figure 6 molecules-28-00377-f006:**
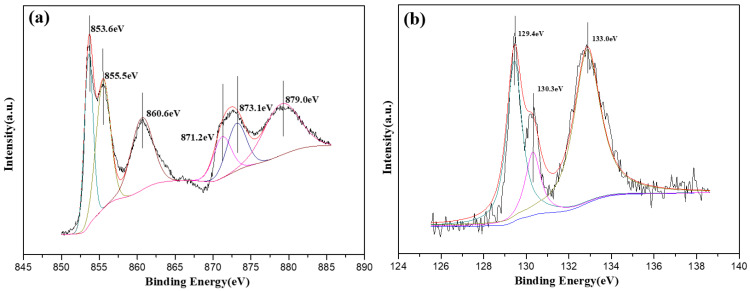
Peak-splitting processing for the alloy coating with heat treatments. (**a**) Peak splitting of Ni at 400 °C, (**b**) peak splitting of P at 200 °C.

**Figure 7 molecules-28-00377-f007:**
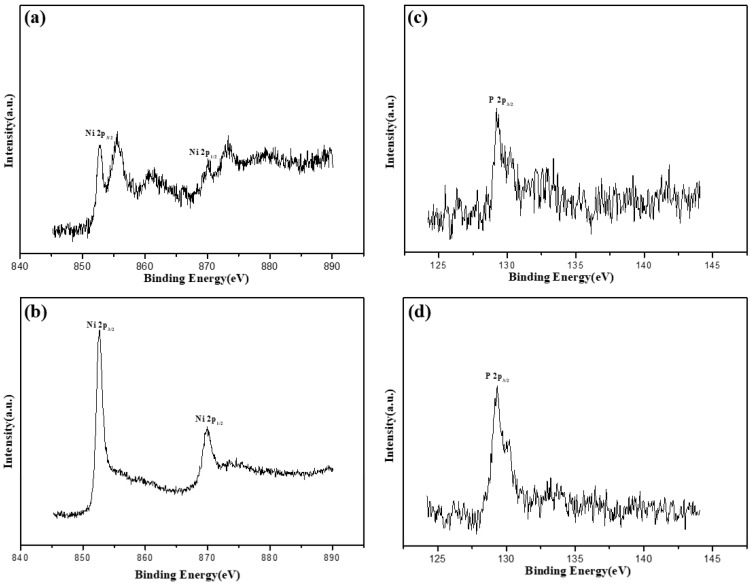
XPS analysis of Ni and P after corrosion of ground-state coating. (**a**) XPS spectrum of Ni, (**b**) XPS spectrum of Ni after etching for 30 s, (**c**) XPS spectrum of P, (**d**) XPS spectrum of P after etching for 30 s.

**Figure 8 molecules-28-00377-f008:**
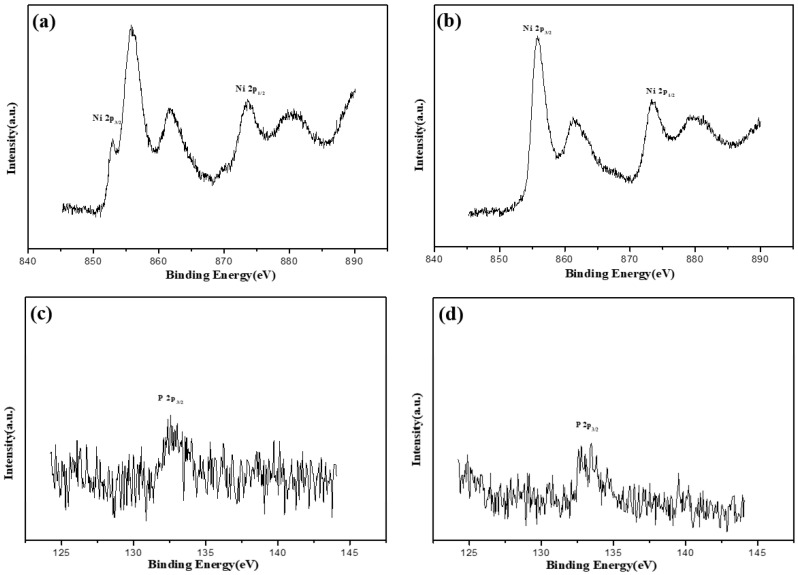
XPS analysis of Ni and P in the corroded coating with heat treatment at 400 °C. (**a**) XPS spectrum of Ni, (**b**) XPS spectrum of Ni element after etching for 30 s, (**c**) XPS spectrum of P, and (**d**) XPS spectrum of P after etching for 30 s.

**Figure 9 molecules-28-00377-f009:**
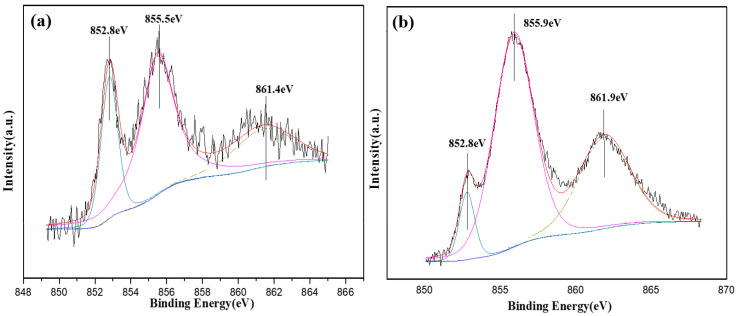
Peak splitting processing for Ni in the corroded alloy coatings at the ground state and with treatment at 400 °C: (**a**) Ni in the coating at ground state, and (**b**) Ni in the coating with treatment at 400 °C.

**Table 1 molecules-28-00377-t001:** Calibration results of Ni and P peaks on the surface of the ground-state alloy coating.

Spectral Line	Binding Energy	Chemical State	Content (wt%)
Ni 2p	852.7	Ni^0^	37.84
	856.1	Ni^2+^ (NiO)	19.08
	870.0	Ni^0^	20.39
	874.0	Ni^2+^ (Ni(OH)_2_)	22.69
P 2p	129.4	P^0^	39.14
	130.4	P^0^	26.45
	132.7	P^+^ (H_2_PO_2^−^_)	34.41

**Table 2 molecules-28-00377-t002:** Calibration results of Ni and P peaks during crystallization.

Spectral Line	Binding Energy	Chemical State	Content (wt%)
Ni 2p (400 °C)	853.6	Ni^2+^ (NiO)	17.82
	855.5	Ni^2+^ (NiO)	21.86
	860.6	Ni^3+^ (Ni_2_O_3_·2H_2_O)	18.86
	871.2	Ni^2+^ (NiO)	8.91
	873.1	Ni^+2^ (Ni(OH)_2_)	10.93
	879.0	Ni^+2^ (NiO)	21.62
P 2p (200 °C)	129.4	P^0^	26.49
	130.3	P^1+^ (H_2_PO_2^−^_)	14.49
	133.0	P^5+^ (H_2_PO_4^−^_)	59.01

**Table 3 molecules-28-00377-t003:** Calibration results of Ni peaks in the alloy coatings at ground state and with heat treatment at 400 °C.

Spectral Line	Binding Energy	Chemical State	Content (wt%)
Ni 2p (ground state)	852.8	Ni^0^	22.3
	855.5	Ni^2+^ (Ni(OH)_2_)	55.8
	861.4	Ni^2+^ (Ni(OH)_2_)	21.9
Ni 2p (400 °C)	852.8	Ni^0^	7.3
	855.9	Ni^2+^ (Ni(OH)_2_)	59.6
	861.9	Ni^2+^ (Ni(OH)_2_)	33.1

## Data Availability

Data available in a publicly accessble repository.

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
