# Peer review of "Exploration of the Corrosion Behavior of Electroless Plated Ni-P Amorphous Alloys via X-ray Photoelectron Spectroscopy"

_molecules, 2023, doi:10.3390/molecules28010377_

Round 1

Reviewer 1 Report

In this work, a Ni-P amorphous alloy was deposited on a low carbon steel substrate via electroless plating. The sample was then crystallized by heating it at a high temperature from 200°C to 500°C 9 in air for 1 h. The experimental results indicate that the diffusion, dissolution, and enrichment of the component elements in the Ni-P alloy are essential during crystallization because the various corrosion behaviors corresponding to Ni and P are directly affected. Overall, this work can be accepted by this journal after minor revision.

1) The language needs great improvement.

2) In Figure 1, the vertical unit should be shown.

3) Some relevant articles related to this work should be cited: Journal of Materials Science: Materials in Electronics, 25: 2611-2617, 2014; Materials Letters, 205: 165-168, 2017; Materials Letters, 196: 205-208, 2017.

4) “This layer combines with H2O via hydrogen bonding to form a physical barrier layer on its outer layer…” -conclusions can be brought in the form of bullet points, rather than a long sentence.-conclusion is not pointed out in the abstract, text or conclusion, and no corresponding data is available to confirm it. This conclusion is not pointed out in the abstract, text or conclusion, and no corresponding data is available to confirm it. It would be better to delete or add corresponding data.

Author Response

Manuscript ID: molecules-2098859

Tittle: The corrosion behavior of electroless plated Ni-P amorphous alloys during crystallization in a NaCl medium via X-ray photoelectron spectroscopy (XPS)

December 12/2022

Dear Professor Reviewer 1,

Thank you very much for your comments concerning our manuscript. Those comments are all valuable and very helpful for revising and improving our paper, as well as the important guiding significance to our researches. We have studied comments carefully and have made correction which we hope meet with approval. Revised portion are marked in red in the paper. The point-by-point responses to the comments are listed in the following pages.

We would like to say that the comments from the referees really enhance the quality of this manuscript. Thanks so much for the comments.

With the best regards,

Zhizhen Li

Response to Reviewer 1 Comments

Point 1: The language needs great improvement.

Responses 1: Thank you for your valuable and thoughtful comments. We have carefully checked and improved the English writing in the revised manuscript.

Point 2: In Figure 1, the vertical unit should be shown.

Responses 2: We are very sorry for our negligence of the vertical axis of figure 1. Figure 1 is the XRD patterns of Ni-P amorphous alloy treated at different temperature heat-treated. It only emphasizes the change characteristics of the XRD patterns of the coatings, but does not affect the integrity of the patterns. The vertical unit is "strength", we have added and shown in figure 1.

Point 3: Some relevant articles related to this work should be cited: Journal of Materials Science: Materials in Electronics, 25: 2611-2617, 2014; Materials Letters, 205: 165-168, 2017; Materials Letters, 196: 205-208, 2017.

Responses 3: Thanks for the comments. As the reviewer suggested, we have carefully reviewed these articles and cited them (Ref. 3, Ref. 4 and Ref. 13) in the revised manuscript. Their conclusions are quoted in our paper.

Point 4: ‘This layer combines with H2O via hydrogen bonding to form a physical barrier layer on its outer layer…”-This conclusion is not pointed out in the abstract, text or conclusion, and no corresponding data is available to confirm it. It would be better to delete or add corresponding data.

Responses 4: Thanks for the comments. As the reviewer suggested, We have re-written this part according to the reviewer’s suggestion, such as “This layer forms a physical barrier and generate a passivation film that greatly inhibits the anodic dissolution of Ni, leading to good corrosion resistance.”

Reviewer 2 Report

The manuscript molecules-2098859 "The corrosion behavior of electroless plated Ni-P amorphous alloys during crystallization in a NaCl medium via X-ray photoelectron spectroscopy (XPS)" presents interesting results about corrosion resistance of Ni-P alloy. The paper presents relevant information. It is well designed and easy to follow. The applied methodology is adequate and the literature cited is highly actualized. It deserves publication in Molecules after Major Revisions. The authors must consider the following comments:

- Change the keywords that are in the title. Provide more innovative keywords.

- The end of the abstract should describe a brief conclusion of the result and importance of the article.

- The end of the introduction section should be rewritten. The authors must justify the reason for the study and what the objectives were.

- The authors discuss the Ni peaks obtained by XRD but do not compare them with the literature. There are some studies on Ni peaks. Add them according to context.

doi: 10.1007/s13204-017-0575-x doi: 10.1016/j.jallcom.2021.159786
doi: 10.1016/j.jwpe.2020.101250
doi: 10.1016/j.mseb.2020.114611

- Figures 1 and 3 must show the x and y axis closed.

- English needs to be improved throughout the manuscript. Avoid using long sentences.

- 33 of the 38 references are in the results section.

The authors need to better endorse the discussion of the results using the literature. As a suggestion, use the same articles previously suggested, as they involve corrosion. Add others that support the results found.

Author Response

Manuscript ID: molecules-2098859

Tittle: The corrosion behavior of electroless plated Ni-P amorphous alloys during crystallization in a NaCl medium via X-ray photoelectron spectroscopy (XPS)

 December 12/2022

 Dear Professor Reviewer 2,

Thank you very much for your comments concerning our manuscript. Those comments are all valuable and very helpful for revising and improving our paper, as well as the important guiding significance to our researches. We have studied comments carefully and have made correction which we hope meet with approval. Revised portion are marked in red in the paper. The point-by-point responses to the comments are listed in the following pages.

We would like to say that the comments from the referees really enhance the quality of this manuscript. Thanks so much for the comments.

With the best regards,

Zhizhen Li

Response to Reviewer 2 Comments

Point 1: Change the keywords that are in the title. Provide more innovative keywords.

Responses 1: Thank you for your valuable and thoughtful comments. We have checked and improved the keywords that are in the revised manuscript.

Point 2: The end of the abstract should describe a brief conclusion of the result and importance of the article.

Responses 2: Thanks for the comments. We have re-written this part according to the reviewer’s suggestion. Please see the revised abstract in the revised manuscript.

Point 3: The end of the introduction section should be rewritten. The authors must justify the reason for the study and what the objectives were.

Responses 3: Thanks for the comments. We have re-written this part according to the reviewer’s suggestion. Please see the revised introduction in the revised manuscript.  

Point 4: The authors discuss the Ni peaks obtained by XRD but do not compare them with the literature. There are some studies on Ni peaks. Add them according to context.

doi: 10.1007/s13204-017-0575-x

doi: 10.1016/j.jallcom.2021.159786

doi: 10.1016/j.jwpe.2020.101250

doi: 10.1016/j.mseb.2020.114611

Responses 4: Thanks for the comments. As the reviewer suggested, we have carefully reviewed these articles and cited them (Ref. 38, Ref. 39, Ref. 43 and Ref. 45) in the revised manuscript. Their conclusions are quoted in our paper.

Point 5: Figures 1 and 3 must show the x and y axis closed.

Responses 5: We are very sorry for our negligence. we have added and shown in Figure 1 and 3.

Point 6: English needs to be improved throughout the manuscript. Avoid using long sentences.

Responses 6: Thank you for your valuable and thoughtful comments. We have carefully checked and improved the English writing in the revised manuscript.

Point 7: 33 of the 38 references are in the results section.The authors need to better endorse the discussion of the results using the literature. As a suggestion, use the same articles previously suggested, as they involve corrosion. Add others that support the results found.

Responses 7: Thank you for your valuable and thoughtful suggestion. We have carefully reviewed these literature. Their conclusions are quoted in our discussion of the results, and their statements better support our conclusions.

Round 2

Reviewer 2 Report

The authors performed all suggested revisions. Therefore the article must be accepted.